# Diff-PCC: Diffusion-based Neural Compression for 3D Point Clouds

## Abstract

Stable diffusion networks have emerged as a groundbreaking development for their ability to produce realistic and detailed visual content. This characteristic renders them ideal decoders, capable of producing high-quality and aesthetically pleasing reconstructions. In this paper, we introduce the first diffusion-based point cloud compression method, dubbed Diff-PCC, to leverage the expressive power of the diffusion model for generative and aesthetically superior decoding. Different from the conventional autoencoder fashion, a dual-space latent representation is devised in this paper, in which a compressor composed of two independent encoding backbones is considered to extract expressive shape latents from distinct latent spaces. At the decoding side, a diffusion-based generator is devised to produce high-quality reconstructions by considering the shape latents as guidance to stochastically denoise the noisy point clouds. Experiments demonstrate that the proposed Diff-PCC achieves state-of-the-art compression performance (e.g., 7.711 dB BD-PSNR gains against the latest G-PCC standard at ultra-low bitrate) while attaining superior subjective quality. Source code will be made publicly available.

## 1 Introduction

Point clouds, composed of numerous discrete points with coordinates (x, y, z) and optional attributes, offer a flexible representation of diverse 3D shapes and are extensively applied in various fields such as autonomous driving [8], game rendering [35], robotics [7], and others. With the rapid advancement of point cloud acquisition technologies and 3D applications, effective point cloud compression techniques have become indispensable to reduce transmission and storage costs.

### 1.1 Background

Prior to the widespread adoption of deep learning techniques, the most prominent traditional point cloud compression methods were the G-PCC [39] and V-PCC [40] proposed by the Moving Picture Experts Group(MPEG). G-PCC compresses point clouds by converting them into a compact tree structure, whereas V-PCC projects point clouds onto a 2D plane for compression. In recent years, numerous deep learning-based methods have been proposed [50, 45, 11, 12, 7, 30, 46, 14, 42], which primarily employ the Variational Autoencoder (VAE) [1, 2] architecture. By learning a prior distribution of the data, the VAE projects the original input into a higher-dimensional latent space, and reconstructs the latent representation effectively using a posterior distribution. However, previous VAE-based point cloud compression architectures still face recognized limitations: 1) Assuming a single Gaussian distribution $N(\mu, \sigma^2)$ in the latent space may prove inadequate to capture the intricate diversity of point cloud shapes, yielding blurry and detail-deficient reconstructions [56, 10]; 2) The Multilayer Perceptron (MLP) based decoders [50, 45, 11, 12, 46] suffer from feature homogenization, which leads to point clustering and detail degradations in the decoded point cloud surfaces, lacking the

Submitted to 38th Conference on Neural Information Processing Systems (NeurIPS 2024). Do not distribute.

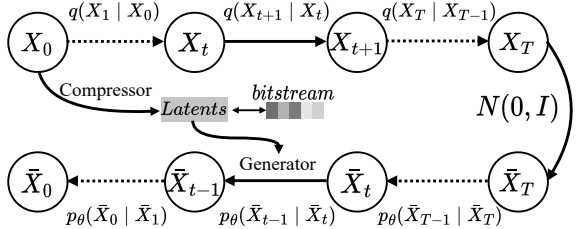

Figure 1: Diff-PCC pipeline. $X_t$ and $\bar{X}_t$ represents the $t$th original point cloud and noisy point cloud, respectively; $p$ refers to the forward process and $q$ refers to the reverse process; $N(0, \boldsymbol{I})$ means the pure noise. Entropy model and arithmetic coding is omitted for a concise explanation.

ability to produce high-quality reconstructions. Recently, Diffusion models (DMs) [5] have attracted considerable attention in the field of generative modeling [34, 48, 41, 19] due to their outstanding performance in generating high-quality samples and adapting to intricate data distributions, thus presenting a novel and exciting opportunity within the domain of neural compression [33, 44, 25]. By generating a more refined and realistic 3D point cloud shape, DMs offer a distinctive approach to reduce the heavy dependence of reconstruction quality on the information loss of bottleneck layers.

## 1.2 Our Approach

Building on the preceding discussion, we introduce Diff-PCC, a novel lossy point cloud compression framework that leverages diffusion models to achieve superior rate-distortion performance with exceptional reconstruction quality. Specifically, to enhance the representation ability of simplistic Gaussian priors in VAEs, this paper devises a dual-space latent representation that employs two independent encoding backbones to extract complementary shape latents from distinct latent spaces. At the decoding side, a diffusion-based generator is devised to produce high-quality reconstructions by considering the shape latents as guidance to stochastically denoise the noisy point clouds. Experiments demonstrate that the proposed Diff-PCC achieves state-of-the-art compression performance (e.g., 7.711 dB BD-PSNR gains against the latest G-PCC standard at ultra-low bitrate) while attaining superior subjective quality.

## 1.3 Contribution

Main contributions of this paper are summarized as follows:

- We propose Diff-PCC, a novel diffusion-based lossy point cloud compression framework. To the best of our knowledge, this study presents *the first* exploration of diffusion-based neural compression for 3D point clouds.
- We introduce a dual-space latent representation to enhance the representation ability of the conventional Gaussian priors in VAEs, enabling the Diff-PCC to extract expressive shape latents and facilitate the following diffusion-based decoding process.
- We devise an effective diffusion-based generator to produce high-quality noises by considering the shape latents as guidance to stochastically denoise the noisy point clouds.

## 2 Related Work

### 2.1 Point Cloud Compression

Classic point cloud compression standards, such as G-PCC, employ octree[29] to compress point cloud geometric information. In recent years, inspired by deep learning methods in point cloud analysis[26, 27] and image compression[1, 2, 22], researchers have turned their attention to learning-based point cloud compression. Currently, point cloud compression methods can be primarily divided into two branches: voxel-based and point-based approaches. Voxel-based methods further branch into

sparse convolution[36, 37, 38, 49, 51, 52] and octree[9, 24, 31]. Among them, sparse convolution derives from 2D-pixel representations but optimizes for voxel sparsity. On the other hand, octree-based methods, utilize tree structures to eliminate redundant voxels, representing only the occupied ones. Point-based methods[11, 50, 45, 46] are draw inspiration from PointNet [26], utilizing symmetric operators (max pooling, average pooling, attention pooling) to handle permutation-invariant point clouds and capture geometric shapes. For compression, different quantization operations categorize point cloud compression into lossy and lossless types. In this paper, we focus on lossy compression to achieve higher compression ratios by sacrificing some precision in the original data.

## 2.2 Diffusion Models for Point Cloud

Recently, diffusion models have ignited the image generation field[58, 17, 32], inspiring researchers to explore their potential in point cloud applications. DPM[20] pioneered the introduction of diffusion models in this domain. Starting from DPM, PVD[57] combines the strengths of point cloud and voxel representations, establishing a baseline based on PVCNN. LION[47] employs two diffusion models to separately learn shape representations in latent space and point representations in 3D space. Dit-3D[23] innovates by integrating transformers into DDPM, directly operating on voxelized point clouds during the denoising process. PDR[21] employs diffusion model twice during the process of generating coarse point clouds and refined point clouds. Point·E[] utilizes three diffusion models for the following processes: text-to-image generation, image-to-point cloud generation, and point cloud upsampling. PointInfinity[13] utilizes cross-attention mechanism to decouple fixed-size shape latent and variable-size position latent, enabling the model to train on low-resolution point clouds while generating high-resolution point clouds during inference. DiffComplete[4] enhances control over the denoising process by incorporating ControlNet[53], achieving new state-of-the-art performances. These advancements demonstrate the promise of DMs in point cloud generation tasks, which motivates our exploring its applicability in point cloud compression. Our research objective is to explore the effective utilization of diffusion models for point cloud compression while preserving its critical structural features.

# 3 Method

Figure 1 illustrates the pipeline of the proposed Diff-PCC, which can also represent the general workflow of diffusion-based neural compression. A concise review for Denoising Diffusion Probabilistic Models (DDPMs) and Neural Network (NN) based point cloud compression is first provided in Sec. 3.1; The proposed Diff-PCC is detailed in Sec. 3.2.

## 3.1 Preliminaries

Denoising Diffusion Probabilistic Models (DDPMs) comprise two Markov chains of length T: diffusion process and denoising process. Diffusion process adds noise to clean data $\boldsymbol{x_0}$, resulting in a series of noisy samples $\{\boldsymbol{x_1}, \boldsymbol{x_2}...\boldsymbol{x_T}\}$. When $T$ is large enough, $x_T \sim N(0, \boldsymbol{I})$. The denoising process is the reverse process, gradually removing the noise added during the diffusion process. We formulate them as follows:

$$q(\boldsymbol{x_1}, \cdots, \boldsymbol{x_T}|\boldsymbol{x_0}) = \prod_{t=1}^{T} q(\boldsymbol{x_t}|\boldsymbol{x_{t-1}}), \text{ where } q(\boldsymbol{x_t}|\boldsymbol{x_{t-1}}) = \mathcal{N}(\boldsymbol{x_t}; \sqrt{1-\beta_t}\boldsymbol{x_{t-1}}, \beta_t \boldsymbol{I}) \qquad (1)$$

$$p_{\boldsymbol{\theta}}(\boldsymbol{x_0}, \cdots, \boldsymbol{x_{T-1}}|\boldsymbol{x_T}) = \prod_{t=1}^{T} p_{\boldsymbol{\theta}}(\boldsymbol{x_{t-1}}|\boldsymbol{x_t}), \text{ where } p_{\boldsymbol{\theta}}(\boldsymbol{x_{t-1}}|\boldsymbol{x_t}) = \mathcal{N}(\boldsymbol{x_{t-1}}; \boldsymbol{\mu_{\theta}}(\boldsymbol{x_t}, t), \sigma_t^2 \boldsymbol{I})$$

$$(2)$$

where $\beta$ is a hyperparameter representing noise level. $t \sim \text{Unif}\{1, \ldots, T\}$ represents time step. Via reparameterization trick, we can sample from $q(\boldsymbol{x_t}|\boldsymbol{x_{t-1}})$ and $p_{\boldsymbol{\theta}}(\boldsymbol{x_{t-1}}|\boldsymbol{x_t})$ as following:

$$x_t = \sqrt{1-\boldsymbol{\beta_t}}x_{t-1} + \sqrt{\boldsymbol{\beta_t}}\boldsymbol{\epsilon} \qquad (3)$$

$$x_{t-1} = \boldsymbol{\mu_{\theta}}(\boldsymbol{x_t}, t) + \sigma_t \boldsymbol{\epsilon} = \frac{1}{\sqrt{\alpha_t}} \left( \boldsymbol{x_t} - \frac{\beta_t}{\sqrt{1-\bar{\alpha}_t}} \boldsymbol{\epsilon_{\theta}}(\boldsymbol{x_t}, t) \right) + \sqrt{\frac{1-\bar{\alpha}_{t-1}}{1-\bar{\alpha}_t}\beta_t}\boldsymbol{\epsilon} \qquad (4)$$

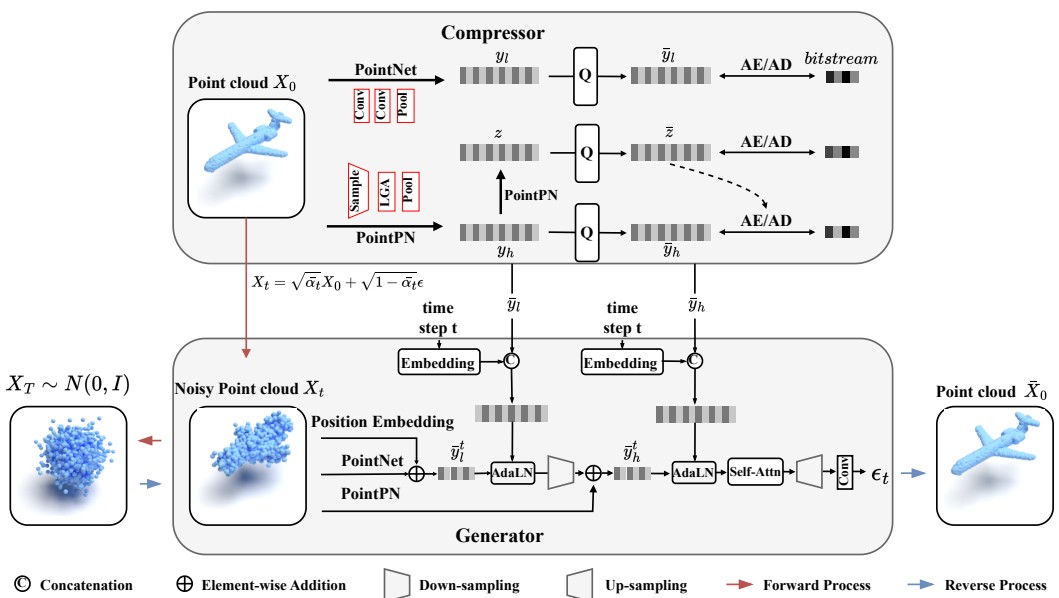

Figure 2: Detailed Structure of the Utilized Compressor and Generator. $y_l$ and $y_h$ refer to the low-frequency shape latent and high-frequency detail latent, respectively; $z$ means hyperprior latent; $Q$ refers to the quantization; AE and AD represents the arithmetic encoding and decoding.

where $\alpha_t = 1 - \beta_t, \bar{\alpha}_t = \prod_{i=1}^{t} \alpha_i$, $\epsilon$ denotes random noise sampled from $\boldsymbol{N}(0, \boldsymbol{I})$. Note that $\epsilon_{\boldsymbol{\theta}}(\boldsymbol{x_t}, t)$ is a neural network used to predict noise during the denoising process, and $\boldsymbol{x_t}$ can be directly sampled via $x_t = \sqrt{\bar{\alpha}_t} x_0 + \sqrt{1 - \bar{\alpha}_t} \boldsymbol{\epsilon}$.

DDPMs train the reverse process by optimizing the model parameters $\theta$ through noise distortion. The loss function $L(\theta, \boldsymbol{x}_0)$ is defined as the expected squared difference between the predicted noise and the actual noise, with the mathematical expression as follows:

$$L(\theta, \boldsymbol{x}_0) = \boldsymbol{E}_{t,\epsilon} ||\epsilon - \epsilon_\theta(\boldsymbol{x_t}, t)||^2 \tag{5}$$

## 3.2 DIFF-PCC

### 3.2.1 Overview

As shown in Fig. 2, two key components, i.e., compressor and generator, are respectively utilized in the diffusion process and denoising process. In Diff-PCC, the diffusion process is identified as the encoding, in which a compressor extracts latents from the point cloud and compresses latents into bitstreams; at the decoding side, the generator accepts the latents as a condition and gradually restoring point cloud shape from noisy samples.

### 3.2.2 Dual-Space Latent Encoding

Several research have demonstrated that a simplistic Gaussian distribution in the latent space may prove inadequate to capture the complex visual signals [56, 3, 6, 10]. Although previous works have proposed to solve these problems using different technologies such as non-gaussian prior [15] or coupling between the prior and the data distribution [10], these techniques may not be able to directly employed on neural compression tasks.

In this paper, a simple yet effective compressor is introduced, which composed of two independent encoding backbones to extract expressive shape latents from distinct latent spaces. Motivated by PointPN [55], which excels in capturing high-frequency 3D point cloud structures characterized by sharp variations, we design a dual-space latent encoding approach that utilizes PointNet to extract low-frequency shape latent and leverages PointPN to characterize complementary latent from high frequency domain. Let $x$ be the original input point cloud, we formulate the above process as:

$$\{y_l, y_h\} = \{E_l(x), E_h(x)\} \tag{6}$$

where $y_l \in \mathbb{R}^{1 \times C}$ and $y_h \in \mathbb{R}^{S \times C}$ represent the low-frequency and high-frequency latent features, respectively; $E_l$ and $E_h$ refer to the PointNet and PointPN backbones, respectively. Next, the quantization process $Q$ is applied on the obtained features $\bar{y}_l$ and $\bar{y}_h$, i.e.,

$$\{\bar{y}_l, \bar{y}_h\} = \{Q(y_l), Q(y_h)\} \tag{7}$$

where function $Q$ refers to the operation of adding uniform noise during training [1] and the rounding operation during test.

Then, fully factorized density model [1] and the hyperprior density model [2] are employed to fit the distribution of quantized features $\bar{y}_l$ and $\bar{y}_h$, respectively. Particularly, the hyperprior density model $p_\varphi(\bar{y}_h)$ can be described as:

$$p_\varphi(\bar{y}_h) = \left( N(\mu, \sigma^2) * \mathcal{U}\left(-\frac{1}{2}, \frac{1}{2}\right) \right)(\bar{y}_h) \tag{8}$$

where $\mathcal{U}\left(-\frac{1}{2}, \frac{1}{2}\right)$ refers to the uniform noise ranging from $-\frac{1}{2}$ to $\frac{1}{2}$; $N(\mu, \sigma^2)$ refers to the normal distribution with expectation $\mu$ and standard deviation $\sigma$, which can be further estimated by a hyperprior encoder $E_{hyper}$ and decoder $D_{hyper}$:

$$(\mu, \sigma^2) = D_{hyper}(\bar{z}) = D_{hyper}(Q(z)) = D_{hyper}(Q(E_{hyper}(y_h))) \tag{9}$$

In this way, a triplet containing quantized low-frequency feature $\bar{y}_l$, quantized high-frequency feature $\bar{y}_h$, and quantized hyperprior $\bar{z}$ will be compressed into three separate streams. Let $p(\cdot)$ and $p_{(\dots)}(\cdot)$ respectively represents the actual distribution and estimated distribution of latent features, then the bitrate $\mathcal{R}$ can be estimated as follows:

$$\mathcal{R} = \mathbb{E}_{\bar{y}_l \sim p(\bar{y}_l)}\left[-\log_2 p_\theta(\bar{y}_l)\right] + \mathbb{E}_{\bar{y}_h \sim p(\bar{y}_h)}\left[-\log_2 p_\varphi(\bar{y}_h)\right] + \mathbb{E}_{\bar{z} \sim p(\bar{z})}\left[-\log_2 p_\phi(\bar{z})\right] \tag{10}$$

### 3.2.3 Diffusion-based Generator

The generator takes noisy point cloud $x_t$ at time $t$ and necessary conditional information $C$ as input. We hope generator to learn positional distribution $F$ of $x_t$ and fully integrate $F$ with $C$ to predict noise $\epsilon_t$ at time $t$. In this paper, we consider all information that could potentially guide the generator as conditional information, including time $t$, class label $l$, noise coefficient $\beta_t$, and decoded latent features ($\bar{y}_l$ and $\bar{y}_h$).

DiffComplete [4] uses ControlNet [54] to achieve refined noise generation. However, the denoiser of DiffComplete is a 3D-Unet, adapted from its 2D version [16]. This structure is not suitable for our method, because we directly deal with points, instead of voxels. We embraced this idea and specially designed a hierarchical feature fusion mechanism to adapt to our method. Note that 3D-Unet can directly downsample features $F$ through 3D convolution with a stride greater than one. It is very complex for point-based methods to achieve equivalent processing. Therefore, we did not replicate the same structure as DiffComplete does, but directly used AdaLN to inject conditional information, formulated as:

$$AdaLN(F_{in}, C) = Norm(F_{in}) \odot Linear(C) + Linear(C) \tag{11}$$

where $F_{in}$ denotes the original features in the Generator and $C$ denotes the condition information.

Now we detail the structure: First, we need to exact the shape latent of noise point cloud $x_t$ and we choose PointNet for structural consistency. However, in the early stages of the denoising process, $x_t$ lacks a regular surface shape for the generator to learn. Therefore, we adopt the suggestion from PDR [23], adding positional encoding to each noise point so that the generator can understand the absolute position of each point in 3D space. Then we inject shape latent $\bar{y}_l$ from the compressor via ADaLN. We formulate the above process as:

$$F_{x_t} = PointNet(x_t) + PE(x_t) \tag{12}$$

$$F'_{xt} = AdaLN(F_{x_t}, C) \tag{13}$$

Next, we need to fuse high-frequency features. We extract the local high-frequency features of $x_t$ using PointPN and add them to $F$ from the previous step, Then we inject the high-frequency features from the compressor via AdaLN. We use K-Nearest Neighbor (KNN) operation to partition locally

and set the number of neighbor points to 8, which allows the generator to learn local details. We formulate the above process as:

$$F^{'} = PointPN(x_t) + FPS(F_{in}) \qquad (14)$$

$$F_{out} = AdaLN(F^{'}, C) \qquad (15)$$

After that, we use the self-attention mechanism to interact with information from different local areas. And through a feature up-sampling module, we generate features for n points. Finally, we output noise through a linear layer. We formulate the above process as:

$$F^{'} = SA(F_{in}) \qquad (16)$$

$$F^{''} = UP(F^{'}) \qquad (17)$$

$$\epsilon_t = Linear(F^{''}) \qquad (18)$$

### 3.2.4 Training Objective

We follow the conventional rate-distortion trade-off as our loss function as follows:

$$\mathcal{L} = \mathcal{D} + \lambda \mathcal{R} \qquad (19)$$

where $\mathcal{D}$ refers to the evaluated distortion; $\mathcal{R}$ represents bitrate as shown in Eq. 10; $\lambda$ serves as the balance the distortion and bitrate. Specifically, a combined form of distortion $\mathcal{D}$ is used in this paper, which considers both intermediate noises ($\epsilon, \bar{\epsilon}$) and global shapes ($x_0, \bar{x}_0$):

$$\mathcal{D} = \mathcal{D}_{MSE}(\epsilon, \bar{\epsilon}) + \gamma \mathcal{D}_{CD}(x_0, \bar{x}_0) \qquad (20)$$

where $\mathcal{D}_{MSE}$ denotes the Mean Squared Error (MSE) distance; $\mathcal{D}_{CD}$ refers to the Chamfer Distance; $\gamma$ means the weighting factor. Here, the overall point cloud shape is additively supervised under the Chamfer Distance $\mathcal{D}_{CD}(x_0, \bar{x}_0)$ to provide a global optimization. The following function is utilized to predict the reconstructed point cloud $\bar{x}_0$ in practice:

$$x_0 = \frac{1}{\sqrt{\bar{\alpha}_t}} \left( x_t - \sqrt{1 - \bar{\alpha}_t} \epsilon_\theta \left( x_t, t, c \right) \right) \qquad (21)$$

where $\bar{\alpha}_t$ means the noise level; $x_t$ refers to the noisy point cloud at time step t; $\epsilon_\theta$ denotes the predicted noise from the generator; $c$ represent the conditional information we inject into the generator.

## 4 Experiments

### 4.1 Experimental Setup

**Datasets** Based on previous work, we used ShapeNet as our training set, sourced from [20]. This dataset contains 51,127 point clouds, across 55 categories, which we allocated in an 8:1:1 ratio for training, validation, and testing. Each point cloud has 15K points, and following the suggestions from [28], we randomly select 2K points from each for training. Additionally, we also used ModelNet10 and ModelNet40 as our test sets, sourced from [43]. These datasets contain 10 categories and 40 categories respectively, totaling 10,582 point clouds. During training and testing, we perform individual normalization on the shape of each point cloud.

**Baselines & Metric** We compare our method with the state-of-the-art non-learning-based method: G-PCC, and the latest learning-based methods from the past two years: IPDAE, PCT-PCC, Following [45, 46], we use point-to-point PSNR to measure the geometric accuracy and the number of bits per point to measure the compression ratio.

**Implementation** Our model is implemented using PyTorch [27] and CompressAI [4], trained on the NVIDIA 4090X GPU (24GB Memory) for 80,000 steps with a batch size of 48. We utilize the Adam optimizer [21] with an initial learning rate of 1e-4 and a decay factor of 0.5 every 30,000 steps, with $\beta_1$ set to 0.9 and $\beta_2$ set to 0.999. Since the positional encoding method requires the dimension (dim) to be a multiple of 6, we designed the bottleneck layer size to be 288. For diffusion, we employ a cosine preset noise parameter, setting the denoising steps T to 200, which is used for both training and testing.

Table 1: Objective comparison using BD-PSNR and BD-Rate metrics. G-PCC serves as the anchor. The best and second-best results are highlighted in **bold** and underlined, respectively.

| Dataset | Metric | G-PCC | IPDAE | PCT-PCC | Diff-PCC |
|---|---|---|---|---|---|
| ShapeNet | BD-Rate (%) | - | -34.594 | -87.563 | **-99.999** |
| | BD-PSNR (dB) | - | +3.518 | +8.651 | **+11.906** |
| ModelNet10 | BD-Rate (%) | - | -35.640 | **-68.899** | -56.910 |
| | BD-PSNR (dB) | - | +4.060 | **+6.333** | +5.876 |
| ModelNet40 | BD-Rate (%) | - | -53.231 | -34.127 | **-56.451** |
| | BD-PSNR (dB) | - | +4.245 | **+6.167** | +5.350 |
| Avg. | BD-Rate (%) | - | -41.550 | -63.530 | **-71.117** |
| | BD-PSNR (dB) | - | +3.941 | +4.384 | **+7.711** |
| Time (s/frame) | Encoding | 0.002 | 0.004 | 0.046 | 0.152 |
| | Decoding | 0.001 | 0.006 | 0.001 | 1.913 |

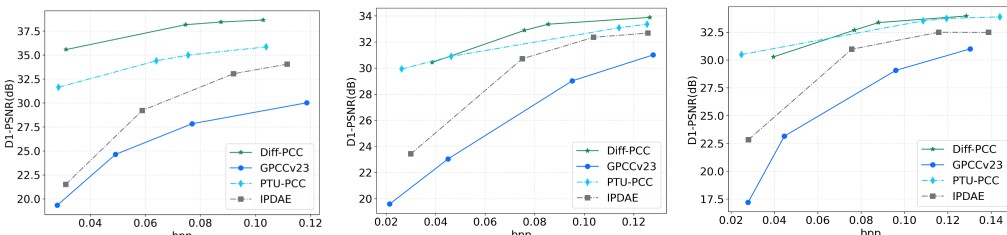

Figure 3: Rate-distortion curves for performance comparison. From left to right: ShapeNet, Model-Net10, and ModelNet40 dataset.

## 4.2 Baseline Comparisons

**Objective Quality Comparison** Table 1 shows the quantitative indicators using BD-Rate and BD-PSNR, and Fig. 3 demonstrates the rate-distortion curves of different methods. It can be seen that, under identical reconstruction quality conditions, our method achieves superior rate-distortion performance, conserving between 56% to 99% of the bitstream compared to G-PCC. At the most minimal bit rates, point ot point PSNR of our proposed method surpasses that of G-PCC by 7.711 dB.

**Subjective Quality Comparison** Fig 4 presents the ground truth and decoded point clouds from different methods. We choose three point cloud:airplane, chair ,and mug. to be tested across a comparable bits per pixel (bpp) range. The comparative analysis reveals that at the lowest code rate, our method preserves the ground truth's shape information to the greatest extent while simultaneously achieving the highest Peak Signal-to-Noise Ratio (PSNR).

## 4.3 Ablation Studies

We conduct ablation studies to examine the impact of key components in the model. Specifically, we investigate the effectiveness of low-frequency features, high-frequency features, and the loss function designed in Sec. 3.2.4. As shown in Table 2, utilizing solely low-frequency features to guide the reconstruction of the diffusion model results in a 20% reduction in the code rate, along with a decrease in the reconstruction quality by 0.397dB. This indicates that high-frequency features play an effective role in guiding the model during the reconstruction process. Conversely, discarding the low-frequency features, which represent the shape of the point cloud, leads to a reduction in the code rate and significantly diminishes the reconstruction quality. Therefore, we argue that the loss of the shape variable is not worth it. Lastly, we ascertain the impact of $\mathcal{D}_{CD}(x_0, \bar{x}_0)$, and the results indicate that this loss marginally increases the bits per point (bpp) while diminishing the reconstruction quality.

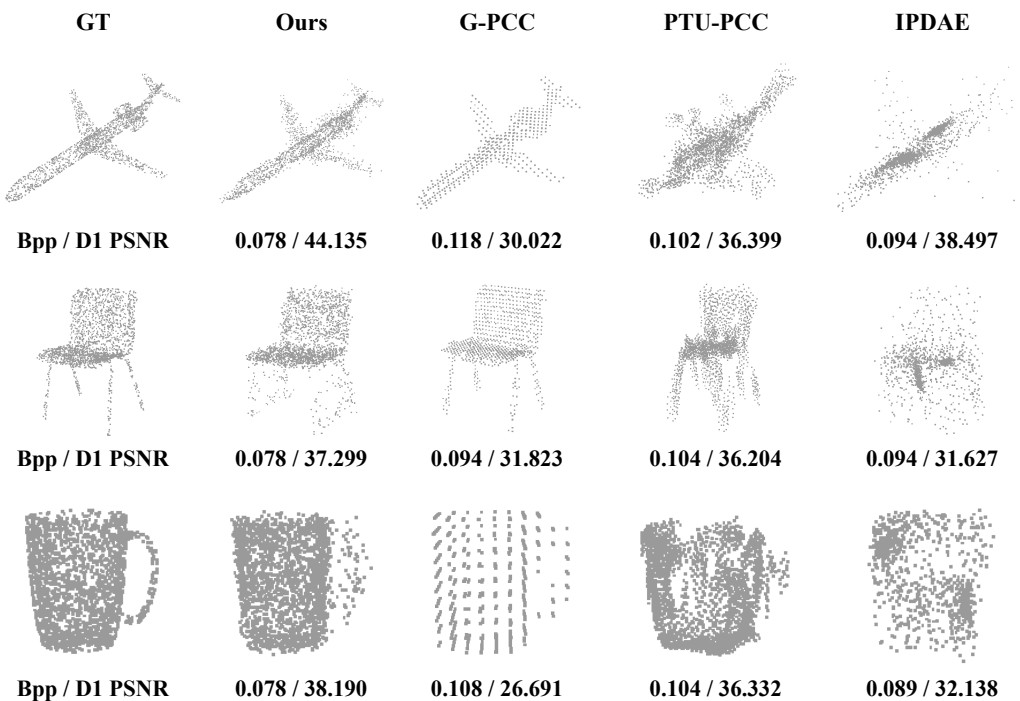

| | GT | Ours | G-PCC | PTU-PCC | IPDAE |
|---|---|---|---|---|---|
| **Bpp / D1 PSNR** | | **0.078 / 44.135** | **0.118 / 30.022** | **0.102 / 36.399** | **0.094 / 38.497** |
| **Bpp / D1 PSNR** | | **0.078 / 37.299** | **0.094 / 31.823** | **0.104 / 36.204** | **0.094 / 31.627** |
| **Bpp / D1 PSNR** | | **0.078 / 38.190** | **0.108 / 26.691** | **0.104 / 36.332** | **0.089 / 32.138** |

Figure 4: Subjective quality comparison. Example point clouds are selected from the ShapeNet dataset, each with 2k points.

Table 2: Ablation study of the proposed method. The original Diff-PCC serves as the anchor.

| $E_l$ backbone | $E_h$ backbone | $\mathcal{D}_{CD}(x_0, \bar{x}_0)$ | BD-PSNR (dB) | BD-Rate (%) |
|---|---|---|---|---|
| ✔ | ✗ | ✔ | -0.397 | -20.637 |
| ✗ | ✔ | ✔ | -2.276 | -16.523 |
| ✔ | ✔ | ✗ | -0.132 | +4.658 |

## 5  Limitations

Although our method has achieved advanced rate distortion performance and excellent visual reconstruction results, there are several limitations that warrant discussion. Firstly, the encoding and decoding time are relatively long, which could potentially be improved by the acceleration techniques employed in several explorations [18, 19]. Secondly, the model is currently limited to compressing small-scale point clouds, and further research is required to enhance its capability to handle large-scale instances.

## 6  Conclusion

We propose a diffusion-based point cloud compression method, dubbed Diff-PCC, to leverage the expressive power of the diffusion model for generative and aesthetically superior decoding. We introduce a dual-space latent representation to enhance the representation ability of the conventional Gaussian priors in VAEs, enabling the Diff-PCC to extract expressive shape latents and facilitate the following diffusion-based decoding process. At the decoding side, an effective diffusion-based generator produces high-quality reconstructions by considering the shape latents as guidance to stochastically denoise the noisy point clouds. The proposed method achieves state-of-the-art compression performance while attaining superior subjective quality. Future works may include reducing the coding complexity and extending to large-scale point cloud instances.

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
