# OpenReview forum: "Diff-PCC: Diffusion-based Neural Compression for 3D Point Clouds"
_NeurIPS.cc/2024/Conference — Submitted to NeurIPS 2024_

### Official Review · Reviewer_HJUZ · 2024-06-23

**Soundness:** 3
**Presentation:** 3
**Contribution:** 3
**Rating:** 5
**Confidence:** 4

**Summary:**

The paper proposes the first diffusion-based point cloud compression method called Diff-PCC.
A dual-space latent representation is devised in this paper, where a compressor composed of two independent encoding backbones is used to extract expressive shape latents from different latent spaces.
At the decoding side, a diffusion-based generator is devised to produce high-quality reconstructions by considering the shape latents as guidance to stochastically denoise the noisy point clouds.
Experiments demonstrate that the proposed Diff-PCC achieves state-of-the-art compression performance (e.g., 7.711 14 dB BD-PSNR gains against the latest G-PCC standard at ultra-low bitrate) while attaining superior subjective quality.

**Strengths:**

Novelty is a strength. To my knowledge, diffusion model is used in point cloud compression for the first time. And the dual-latent design is also novel for learned point cloud compression.

The manuscript is well written and easy to follow. Especially, the author did a good job in introducing related works on image compression, point cloud compression, point cloud analysis and diffusion model.

**Weaknesses:**

More work on diffusion model for data compression could be discussed, like ‘Idempotence and Perceptual Image Compression, ICLR 2024’. In addition, although this paper focuses on point cloud compression, the way of applying diffusion model should be compared with those learned image compression works in the related work part. From my impression, the method in this paper is still novel compared with those learned image compression paper using diffusion model.

More recent learned point cloud compression method [30][14] should be compared in Table 1, Figure 3 and Figure 4, regarding rate distortion and encoding/decoding speed. Besides, only object point cloud is considered currently, large scale point cloud like SemanticKITTI could be compared [30][14].

It is not clear how the speed is measured in Table 1. The hardware and commend line shoud be provided in the supplementary material.

Minor:
L86, Point·E[] is a typo.
[30] and [31] are the same.
L202, the reference should be fixed.
What is the FPS in eq 14? farthest point sampling?

**Questions:**

see the weakness part

**Limitations:**

The limitation is addressed in the manuscript.

---

> ### Author Rebuttal · Authors · 2024-08-06
>
> Dear Reviewer HJUZ,
>
> Thank you for your detailed review and the valuable feedback. We will address your concerns below.
>
>     Q1: More work on diffusion model for data compression could be discussed;The way of applying diffusion model should be compared with those learned image compression works in the related work part.
>
> Thank you for this valuable suggestion. In forthcoming revisions of this article, an introduction to the related research on diffusion models will be further introduced and reviewed
>
>     Q2: More recent learned point cloud compression method (e.g., EHEM and ECM-OPCC) should be compared. Besides, only object point cloud is considered currently, large scale point cloud like SemanticKITTI could be compared.
>
>  • As the mentioned methods, EHEM and ECM-OPCC are not open-sourced yet, it is difficult to directly compare with them. Further, these two existing works require the octree structured to build the entropy model, which is different from ours that can input unstructured and non-voxelized point clouds. Our model Diff-PCC is more generic. We will replicate these two works for comparion in the future.
>
>  • The proposed Diff-PCC, which serves as the first exploration of the cutting-edge 3D diffusion-based compression models, aimning at compressing unstructured relatively small point clouds. Small-scale point clouds are also widely used in real-world applications, such as quick-browsing thumbnails and key points of captured scenes in robotics.
>
>   • To extend its application to large-scale point cloud datasets such as MVUB and MPEG 8i, we can adopt a patch-based approach in this rebuttal. This involves dividing the large point cloud into non-overlapping smaller patches, then sequentially compressing them with Diff-PCC, and finally assembling the reconstructed patches back into a large point cloud.
>
>  • We have displayed the comparison between it and G-PCC in a PDF document. The results indicate that the performance of Diff-PCC is inferior to that of G-PCC. On one hand, the patch-based method may result in the loss of the original semantic information of the whole subjects. On the other hand, it neglects the connections between the patches.
>
> Considering the superior performance demonstrated in the small-scale point cloud samples, future research may include extending this work to compress larget-scale human bodies \& scenes.
>
> ```
>     Q3: It is not clear how the speed is measured in Table 1. The hardwar and commend line shoud be provided in the supplementary material.
> ```
> Thank you for your suggestion. We will update the detailed information in the supplemtray material.
>
> • The hardware and software information is listed below. All our experiments were conducted under the same machine to maintain consistency and reproducibility of the results:
> ```
>      – CPU: Intel(R) Xeon(R) Platinum 8474C @ 2.05GHz
>      – RAM: 128GB DDR4
>      – GPU: NVIDIA GeForce RTX 4090 D
>      – GPU Memory: 24GB
>      – Operating System: Ubuntu 20.04 LTS
>      – CUDA Version: 11.7
>      – cuDNN Version: 8.0.5
>      – Python Version: 3.10.14
>      – PyTorch Version: 2.0.1
> ```
>
> • The command line and test scripts will be provided once the paper is published. Our speed test process follows the conventional user time measurement, which can be briefly described using the pseudo code:
> ```
>      pcl = read_point_cloud(file_path)
>      start_time = time.time() ## mark start time
>      pcl = torch.tensor(pcl) # cpu data to gpu data
>      bytestream = encode(pcl) # encode
>      torch.cuda.synchronize() # wait for CUDA device complete
>      end_time = time.time() ## mark end time
>      print(’Encode Time:’, end_time - start_time)
>      bytestream_to_file(bytestream) # save byte stream
>  ```
>     Q4: Some minor problems.
>
> Thanks for spotting these issues. We will fix them accordingly in forthcoming revision.

---

> > ### Comment · Reviewer_HJUZ · 2024-08-09
> >
> > Thanks for the reply

---

### Official Review · Reviewer_U3Bi · 2024-07-07

**Soundness:** 2
**Presentation:** 3
**Contribution:** 3
**Rating:** 5
**Confidence:** 4

**Summary:**

In this work, the authors propose a diffusion-based point cloud compression framework. Low frequency and high frequency features are extracted via PointNet and PointPN from input point clouds, which are quantized and encoded for compression. During decompression, the quantized features would be decoded to condition a diffusion model to construct the decompressed results. The experiments on 15K points data from ShapeNet, ModelNet10, and ModelNet40 show superiority over compared methods.

**Strengths:**

1. The idea to introduce diffusion models for point cloud compression is different with former works;

2. The paper is easy to follow, while the disgrams are also good;

3. The performances show improvements on sparse point clouds.

**Weaknesses:**

1. The comparison is not convincing enough. Some commonly used compression methods are not compared, while the evaluation is limited to sparse point clouds with relatively simple structures from ShapeNet, ModelNet;

2. The motivation of using diffusion model for compression is questionable. As a sampling-based framework, diffusion models may construct different results during decompression from variant sampled noises each time. I am not so sure if the diffusion model is more appropriate than existing AE or VAE-based frameworks for the compression task, which may need decompression as accurate as possible;
Please check the questions for more details, thanks.

**Questions:**

1. Some popular methods are not compared, including PCGC[1], PCGCv2[2], and 3QNet[3];

[1] Lossy point cloud geometry compression via end-to-end learning

[2] Multiscale point cloud geometry compression

[3] 3qnet: 3d point cloud geometry quantization compression network

2. Only sparse objects on ShapeNet and ModelNet are used for comparison. How is the compression performances on more complex and dense shapes, including 8iVFB[4] or  RWTT[5]? Besides, point clouds with 15K points are too sparse for evaluation as dense points are main targets for compression. Methods mentioned in Question 1 can deal with dense points.

[4] 8i voxelized full bodies-a voxelized point cloud dataset

[5] Real-world textured things: A repository of textured models generated with modern photo-reconstruction tools

3. How do you deal with the uncertainty of diffusion models? The sampling-based generation process may produce different decompressed results between multiple inferences;

4. Could you compare the computational cost, e.g., compression efficiency between different methods?

Some minor problems:

5. In Eq.20, how do you calculate $\bar{x}_0$? As the whole denoising process may be not affordable during training.

6. In Eq.14, is the $F_{in}$ actually $F'_{xt}$ in Eq.13?

**Limitations:**

Yes.

---

> ### Author Rebuttal · Authors · 2024-08-06
>
> Dear Reviewer U3Bi,
>
> Thank you for your detailed review and the valuable feedback. We will
> address your concerns below.
>
>     Q1: Some popular methods are not compared, including PCGC, PCGCv2,and 3QNet.
>
> The proposed Diff-PCC mainly focuses on small-scale point clouds. The mentioned PCGC, PCGCv2 that target at voxlized dense point cloud compression, cannot be directly applied to sparse samples (usually degrates to extremely poor performance). For example, paper [1] has demonstrated the collapse of PCGCv2 on sparse point clouds, which proves that these methods are not on the same track as ours. In constrast, our model Diff-PCC does not require the input point cloud to be voxelized and structured, streamlining the processing pipeline. 3QNet requires external codec Draco to compress the skeleton points of the point cloud, which is not genenrally an end-to-end coding framework as ours.
> We will dissucss the difference in the final version.
>
> [1] IPDAE: Improved Patch-Based Deep Autoencoder for Lossy Point  Cloud Geometry Compression.
>
>     Q2: How is the compression performances on more complex and dense shapes, including 8iVFB or RWTT?
>
> • Diff-PCC is the first exploration of cutting-edge diffusion-based neural models for point cloud compression, currently only targeting at small-scale point cloud compression.To extend its application to large-scale point cloud datasets such as MVUB and MPEG 8i, we can adopt a patch-based approach in this rebuttal. This involves dividing the large point cloud into non-overlapping smaller patches, then sequentially compressing them with Diff-PCC, and finally assembling the reconstructed patches back into a large point cloud.We have displayed the comparison between it and ruled-based G-PCC in a PDF document.The results indicate that the performance of Diff-PCC is  inferior to that of G-PCC. On one hand, the patch-based method may result in the loss of the original semantic information of the whole subjects. On the other hand, it neglects the connections between the patches.
> In summary, by using the patch-based approach, we can apply Diff-PCC to large-scale point cloud compression, and it is worth further exploration.
>
> • Considering the superior performance demonstrated in the small-scale point cloud samples, future research may include extending this work to compress larget-scale human bodies \& scenes.
>
>     Q3: How do you deal with the uncertainty of diffusion models?
>
> Thank you for your detailed review and the valuable feedback.
>
> Despite the issues with the randomness of sampling, we believe that DIFF-PCC, as the first work to introduce DDPM into the field of point cloud compression, holds significant importance and potential for further exploration.
> Thank you for the issue, which is a very valuable research point in our future work.
> Regarding this issue, we have the following two ideas for the future work:
>
> 1. We recognize that DDPM is a branch of the Stochastic Differential Equations(SDE) diffusion model call VP-SDE. SDE can be transformed into Ordinary Differential Equations(ODE) if we eliminate the random terms.Perhaps we could reformulate the problem as a ODE to use the sampling methods in a deterministic way.
>
> 2. DDPM diffuses in the pixel space. For Diff-PCC, the random noise during sampling may directly affect the position of the points in 3D space. Perhaps we could consider the Latent Diffusion Model (LDM) to map the point cloud into latent space.
>
> ```
> Q4: Could you compare the computational cost, e.g., compression efficiency between different methods?
> ```
> We have compared the respective running times of each method in Table 1, with separated encoding and decoding times for better comparison.
>
> ```
> Q5: Problem in Eq.20.
> ```
> When training, we do not obtain $\bar{x}_0$  through T iterations of sampling. Instead, we derive it by reversing the noise addition formula (Eq.21). Although the obtained $x_0$ is relatively coarse, we can still consider it as the final point cloud during training and use it to supervise the training process.
>
> ```
> Q6: Problem in Eq.14.
> ```
> Thank you for pointing out the mistake. In fact, there is no fundamental difference between $F_{in}$ and $ F^{'}_{xt} $ here.We will revise this in the paper accordingly.

---

> > ### Comment · Reviewer_U3Bi · 2024-08-10
> > **Response**
> >
> > Thanks for the rebuttal of the authors. It has addressed some of my concerns. However, I still have some problems for now:
> > (1) I agree that PCGC and PCGCv2 may be unsuitable for sparse point cloud compression. But I hold the opinion that the importance of conducting sparse point clouds compression is not so strong as dense point cloud compression.
> > (2) For your comparisons on dense point clouds, do you normalize the points in different patches? If you do that, then how do you compress the centers and scales of different patches?
> > (3) For the claim about the uncertainty, I think the authors should provide some experiments about the variance of the decompressed results between multiple inferences. Otherwise, it may be difficult to judge if the diffusion framework is appropriate for the task of point cloud compression, due to its instability;

---

> ### Author Response · Authors · 2024-08-14
>
> -Q1:  The importance of conducting sparse point clouds compression is not so strong as dense point cloud compression.
>
> -A1:  Thank you very much for your reply. We agree with your viewpoint.
>     Undoubtedly, dense point clouds are more common in practical applications.
>     The dense point cloud compression offers a wider range of potential applications.
>     However, the original purpose of Diff-PCC was to combine Diffusion with Point Cloud Compression(PCC) to explore the feasibility of this novel technological approach.
>     Although it seems more suitable for sparse point clouds at present, it does not imply that there is no room for improvement in future work.
>     To extend its application to dense point clouds, we are experimenting with the following two methods:
>
> (1) DDPM-based Upsampling: By utilizing the skeleton points and corresponding features of dense point clouds as conditions, we employ diffusion model to upsample large-scale of point, thereby reconstructing dense point clouds.
>
> (2) LDM-based PCC: We map dense point clouds into a latent space with lower dimensions and then diffuse them to reduce computational demands.
>
> Finally, we would like to respectfully point out:
> LiDAR point clouds, as a type of large-scale sparse point cloud, are widely used in the field of autonomous driving and are becoming increasingly important. Voxel models are not adept at handling this kind of data, while point-based models like ours naturally have great potential for processing these point clouds.
>
> -Q2: how do you compress the centers and scales of different patches?
>
> -A2: Yes. We calculate the mean and variance of the point cloud patches and normalize the patches, resulting in arrays with shapes (1, 3) and (1, 1), respectively. The data type is float32, which is expected to occupy 24 bytes for storage.
> In fact, we do not compress the center and scale, but transmit them directly to the decoding end for inverse normalization during this rebuttal period. We will consider to use octree coding to compress the centers.
> Thank you very much for raising the question!
>
> -Q3: About the uncertainty.
>
> -A3:
>   Thank you for your reply.
>   To address your concerns about the uncertainty of the sampling results,
>   In our experiments, we have chosen to fix the random seed, and the code is as follows:
>
>
>     torch.manual_seed(2024)
>     np.random.seed(2024)
>     random.seed(2024)
>
>
> By using this setting, we can ensure that the same random noise is taken during multiple sampling processes, thereby producing stable and consistent decompression results.
>     Despite this, randomness can still lead to some additional issues, such as outliers and rough edges, which are problems we aim to address in the future.
>
>  The above are our views on your question. Thank you very much for your reply. We welcome any insightful suggestions to improve our work.

---

### Official Review · Reviewer_QHjS · 2024-07-12

**Soundness:** 1
**Presentation:** 2
**Contribution:** 1
**Rating:** 2
**Confidence:** 4

**Summary:**

In this paper, they introduce the diffusion-based point  cloud compression method, dubbed Diff-PCC, to leverage the expressive power of the diffusion model for generative and aesthetically superior decoding. They get better performance than G-PCC and two deep learning methods.

**Strengths:**

Encoding point clouds using diffusion models is a good idea. The article is easy to understand.

**Weaknesses:**

Firstly, how do we obtain a point cloud with added noise in the decoder? We have no knowledge of any other information about the original point cloud, except for the information in the bitstream. This will result in the inability to decode.
This manuscript claims to achieve state-of-the-art compression performance, but it only compares with two deep learning methods from the past two years. It does not compare with the most advanced methods such as CNet, SparsePCGC, and so on.

**Questions:**

How do we obtain a point cloud with added noise in the decoder? We have no knowledge of any other information about the original point cloud, except for the information in the bitstream.
How does your method's performance compare to CNet and SparsePCGC?
How does your method perform on datasets such as MPEG 8i and MVUB?

**Limitations:**

The decorder will not work

---

> ### Author Rebuttal · Authors · 2024-08-06
>
> Dear Reviewer QHjS,
>
> Thank you for your detailed review and the valuable feedback. In the following, we address all comments in the review.
>
>     Q1: How to obtain a point cloud with added noise in the decoder?
>
> In a word, the decoder do not need any prior knowledge of the original point cloud. During decoding, a completely random Gaus-
> sian distribution is initialized and then gradually denoised through the information in the bitstream.
>
> Specifically, the decoding process can be described as:
>
> • First, Diff-PCC samples randomly from a Gaussian distribution to obtain a pure noise point cloud $X_T$ with the shape $ (B,N,3) $.
>
> • Then, the generator gradually removes the noise from $X_T$ , generating a series of denoised point clouds  {${X_T, X_{T-1}, ...,X_{1}, X_{0}}$}.
>
> • In this way, the Diff-PCC reconstructs the original point cloud by simulating the reverse process of DDPM in the decoder, starting
> from Gaussian noise $X_T$ , and gradually remove noise.
>
>     Q2: Performance compare with CNet and SparsePCGC on datasets such as MPEG 8i and MVUB.
>
> • Unfortunately, since both CNet (geometry part) and SparsePCGC are not open-sourced yet, it is technically difficult to evaluate these two models on small-scale samples.
>
> • Diff-PCC is the first exploration of cutting-edge diffusion-based neural models for point cloud compression, currently only targeting small-scale point cloud compression (which is also widely used in real-world applications such as thumbnails for quick browsing and key points of captured scenes in robotics).
> However, to extend its application to large-scale point cloud datasets such as MVUB and MPEG 8i, we can adopt a patch-based approach in this rebuttal. This involves dividing the large point cloud into non-overlapping smaller patches, then sequentially compressing them with Diff-PCC, and finally assembling the reconstructed patches back into a large point cloud.
> We have displayed the comparison between it and G-PCC in attached PDF document.
> The results indicate that the performance of Diff-PCC is inferior to that of G-PCC. On one hand, the patch-based method may result in the loss of the original semantic information of the whole subjects. On the other hand, it neglects the connections between the patches.
> In summary, by using the patch-based approach, we can apply Diff-PCC to large-scale point cloud compression, but it is worth further exploration.
>
>
> • Our work Diff-PCC distinguish itslef from previous works in the following aspects:
> (1) We validate the possiblity of applying diffusion probabilistic model into point cloud compression for the first time. (2) Our model is more generic, which support the compression of point clouds of any types, spare or dense, voxelized or non-structured. In contrast, CNet and SparPCGC require the input point cloud to be voxelized.

---

> > ### Comment · Reviewer_QHjS · 2024-08-12
> > **Reply to rebuttal**
> >
> > 1. As writing in the manuscript 'The generator takes noisy point cloud x_t at time t and necessary conditional information C as input.' If x_t is from x_T (a completely random Gaussian distribution), the reverse process has no information about the original point cloud  x_0. In other words, for any original point cloud  x_0, the difference in their x_t is only due to randomness, rather than their own characteristics. So is this reverse process still effective?
> > 2. G-PCC has no relevant configuration files, which will lead to a decrease in performance. Therefore, I think the experiment on ShapeNet and ModelNet is unfair. You said your model is more generic, which support the compression of point clouds of any types, spare or dense, voxelized or non-structured. So please compare with state-of-the-art methods on more common datasets such as MPEG 8i, Ford, KITTI, etc.
> > Compressing point clouds using diffusion models is a good idea, but this work still needs further improvement.  Therefore, we maintain our previous conclusion

---

> ### Comment · Reviewer_QHjS · 2024-08-13
>
> Thank you for addressing the concerns raised in our initial feedback. We acknowledge the effort put into revising the manuscript. However, upon further review, we find that some of the issues highlighted in the weaknesses section are still not adequately resolved.
> 1. As writing in the manuscript. The generator takes noisy point cloud x_t at time t and necessary conditional infommation C as input. lf x_t is from x_T (a completely random Gaussian distribution), the reverse process has no information about the original point cloud x_0. In other words, for any original point cloud x_0, the diference in their x_t is only due to randomness, rather than their own characteristics. So is this reverse process still effective?
> 2. G-PCC has no relevant configuration fles, which will lead to a decrease in performance. Therefore, l think the experiment on ShapeNet and ModelNet is unfair. You said your model is more generic, which support the compression of point clouds of any types, spare or dense, voxelized or non-structured. So you should compare with state-of-the-art methods on more common dafasets such as MPEG 8i, Ford, KlTTl. etc.
> Compressing point clouds using diffusion models is a good idea, but this work still needs further improvement. After thorough consideration, we have decided to maintain our original evaluation and rating of the manuscript.

---

> ### Author Response · Authors · 2024-08-14
>
> We apologize for not thoroughly cleaning up your confusion.Now we will try our best to explain it below:
>
>  -Q1:is this reverse process still effective.
>
>  -A1:In fact, the reverse process contains the information of X_0 because C contains the features extracted from X_0.
> We know that the reverse process starts from X_T (a completely random Gaussian distribution), continuously predicting and removing noise.The generator combines the conditional information C, the time step t, and the noise point cloud X_t to predict the noise.If we do not use C to guide the generator to predict specific noise, the reconstructed point cloud is likely to be very different from the original point cloud. For example, if we input an armchair, without C guidance, it is likely to reconstruct a chair without armrests.However, by following the guidance of C, we can limit the denoising direction and ultimately reconstruct an armchair corresponding to the original point cloud.This process demonstrates the strong generative capabitliy of diffusion model in point cloud generation.We will release our codes and pre-trained model in the near future to demonstrate the reverse process is effective and our decoder can work effectively as illustrated in the paper.
>
> -Q2:Compare with state-of-the-art methods on more common dafasets such as MPEG 8i, Ford, KlTTl. etc.
>
> -A2:
> Thank you for agreeing that “Compressing point clouds using diffusion models is a good idea”,
>     and we  agree that “this work still needs further improvement”.
>     We will  try our best to solve your doubts and the following is our response to this question:
>     By dividing the point cloud into patches, we compared four human point clouds (longdress, loot, redblack, soldier) from MPEG-8i with G-PCC, as shown in the attached PDF document.
>     The configuration file of G-PCC refers to PCGV2.
> We set some parameters as follows:
>
>
>      --positionQuantizationScale=1
>      --trisoupNodeSizeLog2=0
>      --neighbourAvailBoundaryLog2=8
>      --intra_pred_max_node_size_log2=6
>      --inferredDirectCodingMode=0
>      --maxNumQtBtBeforeOt=4
>
> For different point clouds, we select different positionQuantizationScale to control BPP to meet our needs.
> In addition, since SparsePCGC, CNet, and other methods are not fully open-sourced, it is difficult for us to reproduce and we are unable to compare with them currently.
>
> Finally, we welcome any insightful suggestions to improve our work. Thank you very much.

---

### Author Rebuttal · Authors · 2024-08-06

Dear all reviewers,

We thank each of you for generously dedicating your valuable time and
expertise to reviewing our work. We sincerely appreciate your constructive
feedback and are delighted to see the positive comments:

1.Novelty

• Reviewer HJUZ: ”diffusion model is used in point cloud compression for
the first time”;

• Reviewer HJUZ: ”the dual-latent design is also novel for learned point
cloud compression”;

• Reviewer U3Bi: ”introduce diffusion models for point cloud compression is
different with former works”;

• Reviewer QHjS: ”Encoding point clouds using diffusion models is a good
idea”;

2.Writing

• Reviewer HJUZ: ”The manuscript is well written and easy to follow”;

• Reviewer U3Bi: ”The paper is easy to follow, while the disgrams are
also good”;

• Reviewer QHjS: ”The article is easy to understand”;

• Reviewer HJUZ: ”did a good job in introducing related works on
image compression, point cloud compression, point cloud analysis and
diffusion model”;

Detailed responses to each reviewer’s comments are provided in the rebuttal
sections.

---

### Decision · Program_Chairs · 2024-09-25

**Decision:**

Reject

**Comment:**

The reviewers agreed that this paper is not ready for publication yet.
In particular, there was a concern about the evaluation of the method and its stochasticity - how the sample diversity affects reconstruction fidelity and what benefits can it bring. Concerns about comparison with more recent works on more common datasets, were not adequately addressed.